# Complete Resolution of Central Soft Drusen without Geographic Atrophy or Choroidal Neovascularization

**DOI:** 10.3390/jcm11061637

**Published:** 2022-03-16

**Authors:** Rebecca Zeng, Itika Garg, John B. Miller

**Affiliations:** 1Harvard Retinal Imaging Lab, Boston, MA 02114, USA; rebecca_zeng@meei.harvard.edu (R.Z.); itika_garg@meei.harvard.edu (I.G.); 2Retina Service, Massachusetts Eye and Ear, Department of Ophthalmology, Harvard Medical School, Boston, MA 02114, USA

**Keywords:** age-related macular degeneration, drusen resolution, multivitamin supplementation, antioxidant supplementation, geographic atrophy, choroidal neovascularization

## Abstract

The treatment and prevention of dry age-related macular degeneration (AMD) traditionally involve lifestyle modifications and antioxidant supplementation, including the AREDS2 formula. We present a case of a woman with dry AMD in her right eye with several large, confluent central drusen on her exam and optical coherence tomography B-scan. Over the course of a year, the drusen almost completely disappeared, but the retinal layers were preserved without the development of geographic atrophy or choroidal neovascularization. While the exact cause of this phenomenon is unclear, it was thought to be associated with this patient’s strict daily use of numerous dietary supplements. This case highlights the potential in exploring alternative medicine supplements in the treatment of AMD.

## 1. Introduction

Age-related macular degeneration (AMD) is the leading cause of irreversible vision loss in individuals 50 years of age or older in the developed world. Drusen are a clinical hallmark of AMD, as excess drusen lead to inflammation and damage of the retinal pigmentary epithelium (RPE) [1]. The advancement of AMD is marked by the development of geographic atrophy (GA) and/or choroidal neovascularization (CNV) due to the expression of angiogenic cytokines such as vascular endothelial growth factor (VEGF) [2]. Dry AMD is characterized by the presence of drusen, with the development of RPE changes and GA in more advanced stages, while wet AMD is defined by the presence of choroidal neovascularization and subretinal fluid or hemorrhage. Whereas wet AMD is commonly treated with anti-VEGF injections, current management of dry AMD is more conservative, including lifestyle modifications and antioxidant supplementation [1]. Herein, we report a unique case of a woman with dry AMD, who used an extensive list of daily dietary supplements and experienced the dramatic resolution of several large, central soft drusen over the course of a year without the development of GA or CNV.

## 2. Case Presentation

A 64 -year-old Caucasian woman presented for an annual follow-up of dry AMD in her right eye and wet AMD in her left eye. At her last appointment, both eyes were noted to have remained in stable condition since the last Avastin injection of her left eye three years before. At this current appointment, Snellen’s visual acuity (VA) of her right eye improved from 20/50+1 to 20/20−2, while VA in her left eye improved from 20/40+2 to 20/25+1. Of note, she had just undergone a cataract extraction in her right eye three weeks ago. Fundus exam was significant for a dramatic improvement in several large central, soft drusen in the right eye without evidence of subretinal fluid or hemorrhage. Baseline spectral-domain OCT (SD-OCT, SPECTRALIS^®^ Heidelberg), taken a year prior, had shown the presence of large, central drusen with disruption of the outer nuclear layer (ONL) and external limiting membrane (ELM). (Figure 1a). These findings were confirmed on fundus autofluorescence (FAF), which showed a focal area of hyperfluorescence, corresponding to diffuse drusen. (Figure 2) At the current visit, SD-OCT confirmed dramatic regression of the drusen with preservation of the retinal layers, including the ONL. Some hyperreflective material at the level of the ONL can be seen and may represent the retention of material deposits. (Figure 1c) FAF images were not available for this current visit.

Fundus exam of the left eye was stable from the previous visit and showed large and intermediate drusen in a nummular pattern, with collapsed PED and focal atrophy. SD-OCT and FAF confirmed these findings. (Figure 1b,d and Figure 2 e–h)

Upon further questioning, the patient denied leading a particularly healthy lifestyle between her two visits. She revealed that she was a devout Christian and prayed regularly. Lastly, the patient divulged her extensive list of daily dietary supplements (Table 1), prescribed by her long-time orthomolecular specialist.

## 3. Discussion

This is a single case of a patient with near-complete resolution of central large, soft, confluent drusen with preservation of the retinal layers. While drusen resolution is generally regarded as beneficial for the AMD patient, its utility as a clinical trial endpoint is controversial, as not all cases of drusen regression are associated with VA improvement [2]. Drusen regression may occur spontaneously but most often precedes the appearance of new deposits or accompanies progression to late-stage AMD [2]. Progression to GA is more common than progression to choroidal neovascularization [2]. Only ~7% of drusen-only eyes demonstrate significant spontaneous regression without GA or CNV within 24 months [2].

Drusen regression without GA or CNV development has been noted after coincidental rhegmatogenous retinal detachment after vitrectomy with inner limiting membrane peel [3]. This phenomenon was attributed to the elimination of lipophilic material by a large volume of sub-retinal fluid, and stimulation of phagocytosis by surgery, respectively. Treatment with high-dose statins, as well as treatment with low energy photocoagulation and anti-VEGF, has been shown to induce drusen regression without the development of GA and with VA improvement [4,5]. Lastly, subthreshold nanosecond laser was also shown in a pilot study to provoke significant drusen reduction with improved visual function [6]. Although this patient’s VA improvement is confounded by her recent cataract surgery, the resolution of her central confluent drusen on clinical exam and imaging is striking. We were not able to obtain further follow-up from the patient, which may have been useful to confirm the lack of GA and CNV development in a longer time interval.

The exact reason for this patient’s drusen resolution is not completely understood. Aside from recent cataract surgery, the patient had not undergone any other intraocular surgeries, injections, or procedures. She did not take any statins. The dramatic regression of her drusen is especially interesting in the context of her cataract surgery, given that there is some thought that cataract surgery may exacerbate the progression to exudative AMD given its inflammatory effects. This effect is not confirmed, however, as recent reviews have been either inconclusive [7] or unable to find an association between the two [8].

Another possible reason for this phenomenon could be attributed to the patient’s diligent and extensive use of daily dietary supplements. Several of the patient’s supplements, such as lutein, zeaxanthin, Vitamin C and zinc, overlap with the AREDS2 formula, which has been shown to reduce progression to late AMD [9]. Notably, the patient does not take Vitamin E or copper, which are included in the AREDS2 formula [10]. Thus, it is important to consider the role of dietary supplements in the treatment of AMD, given the growing research interest in the role of microbiome and metabolome in this condition [11,12].

Carotenoids, such as lutein, zeaxanthin, and astaxanthin, exert antioxidant, anti-inflammatory and antiapoptotic activities. Thus, supplementation with carotenoids has been shown to reduce CNV risk and improve visual function, including visual acuity, contrast sensitivity and glare disability [13]. Meanwhile, there is evidence that antioxidants, vitamin C, magnesium, and zinc, can slow the progression of AMD [11], but literature regarding their efficacy is mixed [9,14]. Similarly, Vitamin A and D3, also on this patient’s list, have yielded positive but inconsistent studies regarding their efficacy in AMD prevention and treatment [9].

Other supplements on the patient’s list are promising for potential AMD treatments but still need further investigation. Vitamin K2 is implicated in age-related diseases [15]; however, to date, there is no published literature demonstrating its role in AMD treatment. Additionally, alpha-lipoic acid has also been studied in AMD treatment, showing some potential with evidence of antioxidant activity and possible improvement in visual function and quality of life [16,17]. However, its use is limited by gastrointestinal upset issues and failed to show improvement in visual acuity or GA in a recent clinical trial [18]. The last two supplements on the patient’s list—taurine and N-acetylcysteine—have both been shown to have a potential therapeutic effect on human donor RPE cells [19,20]. However, further investigation is needed to ascertain the clinical efficacy of these supplements on AMD treatment.

Beyond this patient’s list, other natural medicine supplements also demonstrate promise. Curcumin has been shown to be neuroprotective [21], with curcumin-based nutritional supplements showing potential to reduce injection burden in patients with wet AMD [22]. Moreover, given increasing evidence of the association between the intestinal microbiota and AMD, further exploration of probiotic treatment in this condition may also be fruitful.

If dietary supplements did contribute significantly to this phenomenon, the unilateral nature of this patient’s improvement should be discussed. Given that AMD is a heterogeneous disease with a variety of phenotypes, with varying sizes and types of drusen [23], it is important to note that the patient’s improvement in the right eye was limited to the central soft, and confluent drusen while some smaller, non-foveal drusen remained. In contrast, the same large and confluent drusen were not apparent in the left eye. In a pilot study of high-dose statins in AMD treatment, Vavvas et al. concluded the medication might have a particular role in a certain subset of AMD patients—namely, those with intermediate-high risk, large drusenoid deposits [5]. Likewise, dietary supplements may play a similar role in benefitting this specific subtype of AMD.

## 4. Conclusions

We describe a rare case of complete resolution of drusen in a patient with dry AMD. The exact treatment modality leading to this extent of dramatic improvement is mysterious and has not been reported before. Considering the morbidity related to the blindness associated with dry AMD, the authors believe this case can be an important step in exploring the efficacy of alternative modalities of treatment as a part of a multi-disciplinary approach via larger sample-sized controlled clinical trials.

## Figures and Tables

**Figure 1 jcm-11-01637-f001:**
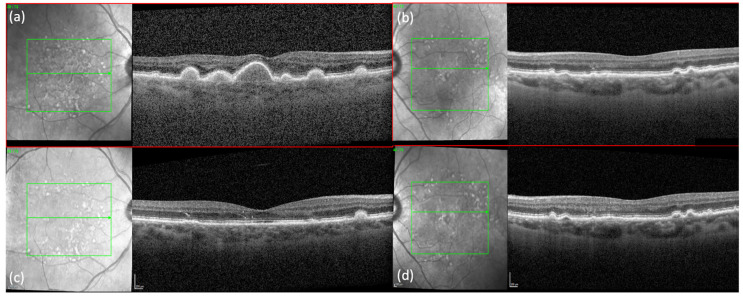
SPECTRALIS^®^ Heidelberg OCT B-scans showed (**a**) presence of several large central, soft, confluent drusen in the right eye and (**b**) collapsed PED and subretinal material without fluid in the left eye. One year later, follow-up B-scan of the (**c**) right eye showed dramatic, complete drusen regression centrally, without geographic atrophy or choroidal neovascularization. (**d**) Corresponding follow-up B-scan of the left eye showed no changes.

**Figure 2 jcm-11-01637-f002:**
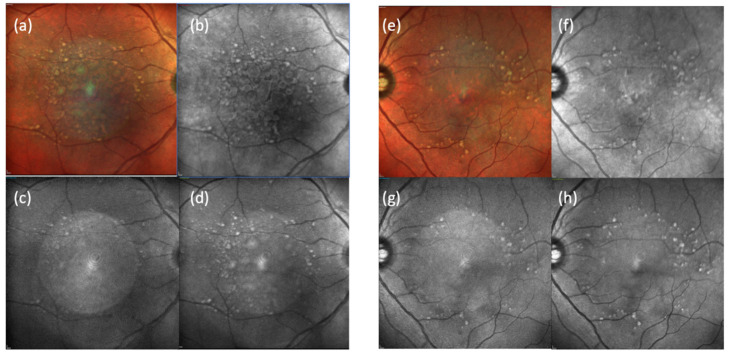
SPECTRALIS^®^ Heidelberg multicolor (**a**,**e**) and fundus autofluorescence with near-infrared (**b**,**f**), blue (**c**,**g**), and green (**d**,**h**) reflectance taken one year prior showed focal areas of hyperfluorescence corresponding to diffuse drusen in both eyes. Multicolor and autofluorescence images were not available for the current visit.

**Table 1 jcm-11-01637-t001:** Daily dietary supplements prescribed to patient by orthomolecular medicine specialist.

Supplements Prescribed for AMD			Other Supplements		
Supplement	Dosage	Frequency	Supplement	Dosage	Frequency
Vitamin D3	5000 IU	4/day	Nutrient 950 (no copper, no iron)	1 cap	2/day
Vitamin A	10,000 IU	1/day	Magtein	667 mg	7/day
Vitamin K2	100 mcg	6/day	Coenzyme Q10	200 mg	3/day
Ascorbyl Palmitate	450 mg	4/day	Synthroid	50 mcg	2/day
(Vitamin C)			Sodium ascorbate	teaspoon	3/4 for bfast, 1/2 for dinner
Astaxanthin	4 mg	2/day	Alpha Lipoic Acid	600 mg	1/day
Lutein	20 mg	4/day	Enteric coated aspirin	81 mg	1/day
Taurine	500 mg	4/day	MSM	1000 mg	4/day
Zeaxanthin	4mg	4/day	Glucosamine sulfate	750 mg	3/day
Magnesium Lethreonate	667 mg	8/day	Quercetin	unknown	
Zinc	30 mg	2/day	Iodoral (Optomax)	12.5 mg	2/day
NAC (Swanson)	600 mg	2/day	Potassium citrate	99 mg	5/day
			Sea salt		1/2 teaspoon per day on food
			Melatonin	10 mg	
			Turmeric		2/day

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
