# Peer review of "Complete Resolution of Central Soft Drusen without Geographic Atrophy or Choroidal Neovascularization"

_jcm, 2022, doi:10.3390/jcm11061637_

Round 1
Reviewer 1 Report
The authors describe a rare case of complete resolution of drusen in a patient with dry AMD and highlight the potential impact of alternative modalities in preventing late AMD.
Please give more details about the case in the abstract. In this case, Spectral domain confirmed drusen regression, but authors should specify if there was preservation of the ONL and ELM at baseline, also at final OCT, there is some hyper reflectivity at the level of the ONL, may be material deposits ? and we need more follow-up to conclude to the spontaneous regression of the drusen without geographic atrophy or neovascularization.
It would be interesting to add autofluorescence images at baseline and on follow up and also is there any result of functional progression of the drusen (microperimetry ? ) It would be interesting to measure choiroidal thickness, is there any change ?
Reviewer 2 Report
The authors reported a rare case of complete resolution of drusen in a patient with dry AMD.
Although this is a very rare and interesting case, there are some questions about their discussions.
The authors seem to think that supplements may have played an important role, but if that is the case, the drusen in the left eye should also improve.
If there is a difference between atrophic AMD and exudative AMD, it should be properly discussed.
It is also thought that cataract surgery may aggravate AMD due to its inflammatory effects. This point should also be taken into consideration. 
The discussion on supplements should be more concise.
Round 2
Reviewer 1 Report
Authors have taken into account the various comments made. the paper could be published in present form
Author Response
Thank you!
Reviewer 2 Report
Many of the points have been appropriately corrected.
I would like to ask the authors for a brief comment on Fig. 2.
What does the circular hyperfluorescence shown in the green (c, g) and blue (d, h) autofluorescence mean? And fundus autofluorescence with near-infra-red are not ‘b’ and ‘d’ but ‘b’ and ‘f’.
